# A Combined Exponential-Power-Law Method for Interconversion between Viscoelastic Functions of Polymers and Polymer-Based Materials

**DOI:** 10.3390/polym12123001

**Published:** 2020-12-16

**Authors:** Vitor Dacol, Elsa Caetano, João R. Correia

**Affiliations:** 1CONSTRUCT (ViBEST), Faculty of Engineering (FEUP), University of Porto, 4200-465 Porto, Portugal; ecaetano@fe.up.pt; 2CERIS, DECivil, IST, University of Lisbon, 1049-001 Lisbon, Portugal; joao.ramoa.correia@tecnico.ulisboa.pt

**Keywords:** viscoelasticity, creep and relaxation, interconversion, stages of creep, mechanical analysis, mechanical characterization, structure–property relationships

## Abstract

Understanding and modeling the viscoelastic behavior of polymers and polymer-based materials for a wide range of quasistatic and high strain rates is of great interest for applications in which they are subjected to mechanical loads over a long time of operation, such as the self-weight or other static loads. The creep compliance and relaxation functions used in the characterization of the mechanical response of linear viscoelastic solids are traditionally determined by conducting two separate experiments—creep tests and relaxation tests. This paper first reviews the steps involved in conducting the interconversion between creep compliance and relaxation modulus in the time domain, illustrating that the relaxation modulus can be obtained from the creep compliance. This enables the determination of the relaxation modulus from the results of creep tests, which can be easily performed in pneumatic equipment or simple compression devices and are less costly than direct relaxation tests. Some existing methods of interconversion between the creep compliance and the relaxation modulus for linear viscoelastic materials are also presented. Then, a new approximate interconversion scheme is introduced using a convenient Laplace transform and an approximated Gamma function to convert the measured creep compliance to the relaxation modulus. To demonstrate the accuracy of the fittings obtained with the method proposed, as well as its ease of implementation and general applicability, different experimental data from the literature are used.

## 1. Introduction

The linear viscoelastic functions that describe the creep and relaxation behavior of polymers and polymer-based materials are mathematically equivalent, and each function contains essentially the same information on the creep and relaxation properties of those materials. Through an appropriate mathematical operation, a linear viscoelastic material function can be converted into another material function.

Polymers and polymer-based materials are extensively used as structural materials in the aerospace and automotive industries, civil engineering [1], and biomechanics. Due to this, a variety of methods applicable to the interconversion of static (creep) and dynamic (relaxation) functions (concerning appropriate experimental data of various polymeric and composite materials) have been investigated over the last few decades to provide a better characterization of those materials [2].

Despite the conceptual simplicity of the viscoelastic properties of polymeric materials, there are operational difficulties in obtaining them from experiments. In this context, interconversion procedures are especially useful due to the difficulty in running constant-strain relaxation tests on stiff materials and the fact that constant-stress creep tests are much easier to carry out [3]; in many cases, constant-stress creep tests can be performed with a simple deadweight system. Concerning relaxation modulus tests, the difficulty involved in the sudden application of a constant strain level was highlighted by Kim [4]. Moreover, it is necessary to adjust the load level during the test to guarantee that the strain is kept constant. Therefore, a method of interconversion becomes an important tool as it allows for more efficient utilization of resources in a rheology laboratory.

In the context of temperature and frequency dependence studies, the interconversion procedure can also be an important tool to help predict the behavior of polymer-based materials [5].

Different methods have been proposed to solve the interconversion expression of linear viscoelasticity. In this context, reference is made to an extensive theoretical treatment of the subject given in [6,7]. Other studies, such as [8,9,10], have focused on numerical approaches using Prony series or power laws to characterize the creep and relaxation moduli of polymeric materials.

When studying the mechanical behavior of viscoelastic materials, it is usually assumed that the strains and the stresses are deterministic functions. However, Wang et al. [11] highlighted the importance of considering the uncertainty in the propagation of the frequency response for the robust design of viscoelastic damping structures. According to De Lima et al. [12], among the various methods devised for uncertainty modeling, the stochastic finite element method has received major attention, as it is well adapted for application to complex engineering systems. Despite their potential importance, the experimental quantification of uncertainties and the corresponding stochastic modeling are not addressed in the present study.

In the first part of this paper, a general theoretical background of the viscoelastic behavior of polymeric composites is provided, and a brief review of interconversion methods is presented and discussed. In the second part, based on the works of Crevecoeur [13,14] and the study conducted in [15], this paper proposes a new interconversion procedure, which involves the combination of Laplace transform theory with an approximated Gamma function to convert the measured creep compliance to the relaxation modulus. The main goal of this paper is to demonstrate (i) the mathematical procedure to interconversion from creep to relaxation functions and (ii) the good fitting for creep and relaxation data provided by the proposed methodology.

## 2. Theoretical Background

### 2.1. Constitutive Relations

The creep and relaxation properties of polymeric and polymer-based materials allow for prediction of their response when subjected to a step function of stress or strain as a function of time. To predict the response of a material to any history of stress or strain as a function of time, the constitutive integrals of linear viscoelasticity must be used. The creep function is defined as a strain variation as a function of time under constant stress. In the linear range, the Boltzmann superposition principle is valid and expressed as
(1)$$\varepsilon \left(t\right)=\int_{0}^{t}D\left(t-\tau \right)\frac{\partial \sigma \left(\tau \right)}{\partial \tau }d\tau$$
where ε(t) is the strain, t is the time, σ is the stress, and D is the creep compliance function.

If the roles of stress and strain are interchanged and the above arguments are repeated, a complementary relationship is obtained:(2a)$$\sigma \left(t\right)=\int_{0}^{t}E\left(t-\tau \right)\frac{\partial \varepsilon \left(\tau \right)}{\partial \tau }d\tau$$
where E is the relaxation modulus function.

Thus, if the response of a material to a step function of stress or strain has been determined experimentally, the response to any load history can be found for analysis or design.

It must be noted that the relaxation and creep functions are not arbitrary functions, and the restrictions on the form of these functions have been derived based on physical principles [16].

### 2.2. The Relation between Creep and Relaxation in the Time Domain

A simple relationship between the creep function D(t) and the relaxation function E(t) is convenient to simplify the viscoelastic material characterization. Both functions appear in the Boltzmann superposition integrals (1) and (2), described above.

The two integral expressions may be manipulated with the aid of integral transforms, such as the Laplace transform
(2b)$$\bar{f}\left(s\right)=\int_{0}^{\infty }f\left(t\right)e^{-st}dt,\;\;t>0$$
where s is the transform variable, $e^{-st}$ is the transformed kernel function, and $\bar{f}$ is the rational function of s.

By integration of (1) and (2) and using the convolution theorem, the following relations are obtained:(3)$$\tilde{\sigma }\left(s\right)=s\bar{E}\left(s\right)\bar{\varepsilon }\left(s\right)$$
(4)$$\tilde{\varepsilon }\left(s\right)=s\bar{D}\left(s\right)\bar{\sigma }\left(s\right)$$
where $\bar{\sigma }\;$is the stress operator in the Laplace domain, $\bar{\varepsilon }$ is the strain operator in the Laplace domain, and $\bar{E}\;$and $\bar{D}$ are the rational functions of s.

By appropriate mathematical operations, (3) and (4) are manipulated, yielding
(5)$$\bar{E}\left(s\right)\bar{D}\left(s\right)=\frac{1}{s^{2}}$$

The relation (5) between the Laplace-transformed relaxation and creep functions is useful in directly evaluating one transformed function when the other function is known at a particular value of the transform parameter [3].

Applying the inverse of the Laplace transform to (5) and using the convolution theorem and the Laplace operator relation $\bar{f}\left(t\right)=1/s^{2}$ yields
(6)$$\int_{0}^{t}E\left(t-\tau \right)D\left(\tau \right)d\tau =t$$
or
(7)$$\int_{0}^{t}D\left(t-\tau \right)E\left(\tau \right)d\tau =t$$

Further, differentiating (6) and setting the time to zero (t=0), the initial condition between D(t) and E(t) yields
(8)D(0)E(0)=1

The convolution integrals relating the creep compliance and the relaxation modulus (Expressions (6) and (7)) are the Volterra expressions of the first kind.

## 3. Interconversion Methods: A Brief Review

Several approximate methods of interconversion between transient relaxation and creep functions are available in the literature. Most of them are based on power laws or exponential functions. In this section, some of these methods are presented and discussed.

### 3.1. Power-Law-Based Interrelationship by Leaderman (1958)

For many linear viscoelastic materials, the relaxation modulus and the creep compliance function are approximately represented by simple power laws over their transition zones. The simplest form of a power law is the pure power-law (PPL), which is described by the following time loading description for the relaxation modulus [17]:(9)$$E\left(t\right)=E_{1}t^{-n}$$
and the following time loading description for the creep compliance:(10)$$D\left(t\right)=D_{1}t^{n}$$
where $E_{1},\;D_{1},\;\mathrm{and}\;n\;$are material constants.

From (9) and (10), by Laplace transform operations, the following relation occurs:(11)$$E\left(t\right)D\left(t\right)=\frac{\sin\left(n\pi \right)}{n\pi }$$

The main advantage of the power-law-based approach is its simplicity. As is well known, the power-law functions are graphically represented by straight lines on log–log scales, and the exponent n is identified as *the material regression coefficient* representing the creep or relaxation rate.

### 3.2. Log–Log Slope-Based Interrelationship by Park and Kim (1999)

Park and Kim [5] introduced an approximate interconversion method that uses the local log–log slope of the source function and is based on the concept of equivalent time determined by rescaling the physical time. The rescaling factor, α, which can be interpreted as a shift factor on a logarithmic time axis, is defined as the local slope of the source function on log–log scales.

From (9) and (10), one may obtain the following expressions relating D(t) and E(t):(12)$$D\left(t\right)=\frac{1}{E\left(\alpha t\right)}$$
(13)$$E\left(t\right)=\frac{1}{D\left(\frac{t}{\alpha }\right)}$$

The rescaling factor, α, is then given by
(14)$$\alpha ={\left(\frac{\sin\left(n\pi \right)}{n\pi }\right)}^{\frac{1}{n}}$$

This method produces good results when the source functions are characterized by broadband and smooth behavior on doubly logarithmic scales [3].

### 3.3. Exponential-Based Interrelationship by Schapery and Park (1999)

The use of the Dirichlet series, also known as the Prony series, is a popular method to represent the relaxation function, mainly due to its ability to describe a wide range of viscoelastic responses, associated with the relative simplicity of its exponential basis functions and rugged computational efficiency. In the Dirichlet model approach, the generalized Maxwell model is used as an analogy to represent the creep and relaxation functions.

The generalized Maxwell model (or Wiechert model) consists of a spring and n Maxwell elements connected in parallel. The relaxation modulus derived from this model is given by
(15)$$E\left(t\right)=E_{0}+\overset{m}{\underset{i=1}{\sum }}E_{i}e^{-(t/\tau_{i})}$$
where $E_{0}\;$is the equilibrium modulus, $E_{i}\;$is the relaxation strength, and $\tau_{i}\;$is the delay time.

The analytical description of the creep compliance function of the generalized Maxwell model is expressed by the sum of exponential terms
(16)$$D\left(t\right)=D_{0}+\overset{m}{\underset{i=1}{\sum }}D_{i}\left[1-e^{-(t/\tau_{i})}\right]$$
where $D_{0}$ is the glassy compliance and $D_{i}$ is the retardation strength.

In this scheme, because both the source and the target functions are represented in the Prony series, the complete target function can be determined by solving a system of linear algebraic expressions given by [3]
(17)[A]{E}=[B]
where
(18)[A]=∑k=1p∑i=1m[D1(e−tkρi−1)+∑j=1nDjτjρi−τj(e−tkρi−e−tkτj)]
and
(19)[B]=1−∑k=1pE1[D1+∑j=1nDj(1−e−tkτj)]

The main inconvenience of the Dirichlet series adjustment is the potential problem of numerical instability resulting from the generation of negative dependent terms.

## 4. The Convolution Theorem

It is well known that the convolution theorem can be used to solve integrals and integral–differential expressions. The basic mathematical definition of convolution is the integral over all space of one function at x multiplied by another function at u-x, concerning to x, where x can represent anything, including time, frequency, or even a three-dimensional space, depending on the application [18].


*“The convolution theorem for*
s
*-transforms states that for any (real or) complex causal signals*
x
*and*
y
*, the convolution in the time domain is equivalent to a multiplication in the*
s
*domain.”*


Given the functions of time t,
f(t) and g(t), the convolution (*) of f(t) and g(t) is a function of t defined by
(20)$$\left(f*g\right)\left(t\right)=\overset{\infty }{\underset{-\infty }{\int }}f\left(t-\tau \right)g\left(\tau \right)d\tau$$

Applying the ***s***-Laplace transform to the second member of (20) and changing the order of the integrals yields
(21)$$\bar{f}\left\{\left(f*g\right)\left(t\right)\right\}=\int_{0}^{\infty }\int_{0}^{t}f\left(t-\tau \right)g\left(\tau \right)e^{-st}d\tau dt$$

Finally, changing variables in the inner integral (substituting υ=t−τ, dν=dt (τ a constant)) (21) yields
(22)$$\bar{f}\left\{\left(f*g\right)\left(t\right)\right\}=F\left(s\right)G\left(s\right)$$

## 5. Exponential-Power-Law (EPL) Interconversion

It has been shown in [19] that the creep curve of polymers and polymer-based materials can be well described by the following expression:(23)$$\varepsilon \left(t\right)=k\cdot t^{\beta }\cdot e^{\frac{t}{t_{i}}}$$
where ε(t) is the calculated strain, β is a constant related to strain hardening (β<1 and α≪β), k is a constant related to the main stress influence, and $t_{i}$ is the instability time.

Indeed, the advantages of (21) in providing a good fitting to a creep curve are summarized as follows: (i) it is a very simple approximation, as only three parameters have to be determined to fit the given creep data $\left(k,\;t_{i},\;\mathrm{and}\;\beta \right)$; (ii) it provides a good quality of extrapolations because it does not involve addition or subtraction operations; (iii) it combines the three stages of creep; and (iv) it is generally applicable.

As previously shown by the authors [15], the main advantage of using the exponential-power-law (EPL) method as given by expression (21) is its ability to cover the full path of the creep behavior and its capacity to estimate the respective transition points between stages.

Conversion between the creep compliance, fitted with (21), and the relaxation modulus may be required to estimate the fatigue resistance of composite structures; to that end, the relaxation behavior and the corresponding transition points should be known.

### Laplace Transform of EPL

Using the Laplace transform properties summarized in Table 1, each term of the EPL expression (21) can be transformed as
(24)$$\bar{f}\left\{e^{\frac{t}{t_{i}}}\right\}=\frac{1}{s-\frac{1}{t_{i}}}$$
(25)$$\bar{f}\left\{t^{\beta }\right\}=\frac{\beta !}{s^{\beta +1}}$$

After mathematical manipulations, the EPL expression (21), in ℒ domain, is transformed into
(26)$$\bar{\varepsilon }\left(s\right)=k\cdot \frac{\beta !}{s^{\beta +2}+t_{i}s^{\beta +1}}$$

From (22), the creep compliance can be written as
(27)$$\bar{D}\left(s\right)=D_{0}\cdot \frac{\beta !}{s^{\beta +2}+t_{i}s^{\beta +1}}$$
where $D_{0}\;$is the compliance in the glassy state, given by D0=kσ.

From the relationship given by (5), the relaxation modulus is defined in the Laplace domain as
(28)$$\bar{E}\left(s\right)=D_{0}{}^{-1}\cdot \frac{s^{\beta +2}+t_{i}s^{\beta +1}}{s^{2}\beta !}\;$$

After necessary simplifications, it follows that
(29)$$\bar{E}\left(s\right)=\frac{\sigma }{k}\cdot \frac{s^{\beta }+t_{i}s^{\beta -1}}{\beta !}$$

Applying the inverse of the Laplace transform to (29), one finds
(30)$$E\left(t\right)=\frac{\sigma }{k\;t^{\beta }}\left[\frac{t^{-1}}{\beta \Gamma \left(\beta \right)\Gamma \left(-\beta \right)}-\frac{t_{i}}{\beta \Gamma \left(\beta \right)\Gamma \left(1-\beta \right)}\right]$$

Here, the operator Γ is the Euler Gamma function.

The first definition of the Γ-function resulted from the idea of extending the positive integer factorial n!=n(n−1)(n−2)… to real numbers [20].

The Γ-function is an extension of the factorial function with an argument shifted by one, to real numbers and the complex numbers, as follows:(31)$$\Gamma \left(z\right)=\left(z-1\right)!=\overset{\infty }{\underset{0}{\int }}t^{z-1}e^{-t}dt,\;\;ℜ\left(z\right)>0$$

The Γ-function defined above can be integrated by parts to show the recurrence or functional relation, yielding
(32)z!=Γ(z+1)=zΓ(z)

The Γ-function does not have a closed-form solution and needs to be approximated.

The Stirling approximation is the most well-known and cited approximation of the Γ-function [20] and is defined as
(33)$$z!=\Gamma \left(z+1\right)\;\sim \;{\left(\frac{z}{e}\right)}^{z}\cdot {\left(2\pi z\right)}^{\frac{1}{2}}$$

Here, the signal “~” means that the two quantities are asymptotic, i.e., their ratio tends to one as z tends to infinity. Despite its effectiveness, the Stirling formula is not the most accurate, especially for values in the range 0<z<1.

Burnside’s asymptotic formula for factorial z provides a more efficient estimation of the Γ-function compared to (28). Furthermore, Gosper’s formula [21] provides a slightly more accurate estimation of the Γ-function than Stirling’s formula.

An important property of the Γ-function is Euler’s reflection formula, where, for all non-integer z, one finds
(34)$$\Gamma \left(z\right)\Gamma \left(1-z\right)=-z\Gamma \left(-z\right)\Gamma \left(z\right)=\left(-z\right)!\Gamma \left(z\right)⋍\frac{\pi }{\sin\pi z}$$

As shown in [22], a Γ-function of negative values may be written as follows:(35)$$\Gamma \left(-z\right)=\frac{\Gamma \left(-z+1\right)}{-z}=\frac{\Gamma \left(1-z\right)}{-z}$$

Applying the numerical approximation discussed above, the relaxation modulus of the EPL creep function given by (25) becomes
(36)$$E\left(t\right)=\frac{\sigma }{k\;t^{\beta }}\cdot \frac{\sin\left(\pi \beta \right)}{\pi }\cdot \left(\frac{t_{i}}{\beta }+\frac{1}{t}\right)$$

Based on the creep test data provided in [23] for specimen T28-S15 (a glass-fiber-reinforced polymer (GFRP) laminate tested in three-point bending), Figure 1 presents a graphical representation of the time variation of the expression (25) and the approximation (31)—it can be seen that these functions present a perfect match.

## 6. Application

This section presents the application of the method described above to the creep and stress relaxation tests presented by Garrido et al. [23], Sorvari et al. [24], and Hernández-Jiménez et al. [25].

### 6.1. Creep Compliance and Relaxation Modulus Fitted from Garrido et al. (2015)

#### 6.1.1. Creep Compliance Fitting

The GFRP laminates used in [23] were manufactured by vacuum infusion, using E-glass fiber rovings and mats embedded in orthophthalic polyester resin. The fiber layup followed an [0/0/30/ - 30/90/0]_S_ arrangement, resulting in a laminate with a nominal thickness of 7 mm, a nominal fiber volume fraction of 45%, and a fiber weight of 9000 g/m^2^. The glass transition temperature of the material was measured as $T_{g}=69.3\;{}^{\circ}C$. Further mechanical properties can be found in the referenced paper.

The flexural creep behavior was measured and considered to be representative of the overall bending creep behavior by providing a combination of the compressive and tensile creep responses of the laminate. The creep experiments were carried out at 20, 24, and 28 °C, for stress levels corresponding to 15%, 25%, and 35% of the laminate’s flexural strength, with test durations ranging between 1000 and 2200 h. The specimens were labeled as “$T_{i}-S_{j}$”, i.e., specimens tested at temperature *i* and stress level *j*.

As the main goal of this paper is to verify the fitting accuracy of the EPL expression and its interconversion method, only data from the 28 °C experiment are used.

The first step is to fit the creep curve to three levels of stress. This procedure is necessary to quantify the influence of stress on the material creep behavior. From the data spectra, the EPL expression coefficients are fitted and shown in Table 2.

Figure 2, Figure 3 and Figure 4 show the fitting of the full path of creep from the EPL expression to the three specimens tested at 28 °C: T28-S15, T28-S25, and T28-S35.

As described in [15], the points represented in these figures are as follows:

$t_{1ary}$ is the transition point between transient and steady creep;$t_{2ary}$ is the inflection point of the creep curve;$t_{3ary}$ is the second transition point (flow time);$t_{i}\;$is the instability time;$t_{m}\;$is the maximum possible lifetime.

Based on $D_{0}$, the compliance in the glassy state is given by D0=kσ. Table 3 summarizes the values of $D_{0}$ and the parameters used to obtain the creep compliance for each stress level.

#### 6.1.2. Relaxation Modulus Prediction

Since the creep tests were conducted under constant stress, the relaxation modulus can be expressed in terms of the EPL constants using the EPL relaxation expression (31), leading to the parameters summarized in Table 4.

Figure 5 shows the full path of the creep compliance and the relaxation modulus by EPL expressions.

As shown in Figure 5, both the creep compliance and the relaxation modulus present stress dependence. According to [26], this dependence becomes more critical as the temperature approaches the glass transition temperature.

As reported above, the creep compliance and the relaxation modulus are not each other´s reciprocals but they are instead related. Figure 6 shows the relationship between these viscoelastic properties.

### 6.2. Relaxation Modulus Fitted from Sorvari et al. (2006)

Sorvari et al. [24] presented numerical procedures to evaluate the creep compliance from relaxation tests with nonideal loading and noise in simulated experimental data. From several ramp tests, the relaxation modulus was determined under the step strain assumption. The interconversion methods were numerically tested using synthetic data. The following procedure was applied to simulate the determination of creep conformity from the relaxation modulus: (i) the virtual relaxation test was made, from which the stress was determined; (ii) the relaxation modulus was computed; and (iii) the creep compliance was determined from a convenient method (for more details, see [24]). According to the authors, the procedure described above was carried out in different simulation cases with varying material parameters and noise.

The relaxation modulus was chosen to be
(37)$$G\left(t\right)=G_{0}+G_{1}e^{-t/\lambda }$$
where $G_{0}\;$is the initial shear modulus, $G_{1}\;$is the constant shear modulus, and λ is the relaxation time.

From the given simulation, the coefficients of (32) were $G_{0}=0.4\;\mathrm{MPa}$, $G_{1}=0.5\;\mathrm{MPa}$, and λ=1 to 100 s. In this paper, the relaxation modulus for λ=2 s and λ=4 s is fitted. The average relative error was 1.53% for λ=2 s and 1.78% for λ=4 s. In both cases, the r-Pearson coefficient was higher than 0.95.

The relaxation modulus is expressed in terms of the EPL constants using the EPL relaxation expression (31), leading to the parameters summarized in Table 5.

Figure 7 and Figure 8 show the full path of the relaxation modulus obtained from the EPL expressions.

### 6.3. Relaxation Modulus Fitted from Hernández-Jiménez et al. (2002)

Hernández-Jiménez et al. [25] presented a study of stress relaxation in samples of methylmethacrylate (PMMA) and polytetrafluorethylene (PTFE) polymers; in this study, it was pointed out that there is not only one time of relaxation as predicted by the classic Maxwell model, but rather two distributions of relaxation time.

Two different types of commercial-grade polymer samples from Dupont were used. Standard test samples of dimensions 100 mm × 25 mm × 3 mm for PTFE and 100 mm × 25 mm × 4 mm for PMMA were used and deformed under tension at a crosshead speed of 3 mm/min. In the stress relaxation test, when the deformation was 5%, the testing machine crosshead was stopped and kept in a constant position, and the values of the applied load, F(t), were registered over time. From F(t), the respective stress, σ(*t*), was obtained by dividing it by the normal cross-section of the sample. Finally, the time relaxation modulus function was obtained by dividing the calculated stress by the constant strain of the sample, $\varepsilon_{0}=0.05$.

The relaxation modulus is expressed in terms of the EPL constants using the EPL relaxation expression (31), leading to the parameters summarized in Table 6.

Figure 9 shows the relaxation modulus fitted by the EPL expressions. The average relative error was 1.85%, and the r-Pearson coefficient was 0.95.

## 7. Accuracy and Validation

To perform an accuracy test and further validate the EPL fitting and its interconversion, sampled points from the relaxation curves identified in [27] were used, corresponding to cylindrical asphalt mixture specimens at three aging conditions (three replicates for each experimental combination). Direct tension relaxation modulus tests were conducted at 21 °C using the trapezoidal loading pattern at a low level of strain.

The standard sigmoid (SS) and generalized logistic sigmoid (GLS) fitting expressions are
(38)$$E\left(t\right)=18.99+\frac{623.61}{1+e^{2.53+1.017\;(\log t)}}$$
(39)$$E\left(t\right)=21,32+\frac{649.63}{{\left[1+0.67e^{2.02+0.83\;(\log t)}\right]}^{1.50}}$$

It must be noted that these functions may not provide exact representations of experimental data. However, for a given set of constants, they are exact and thus provide a means for checking the accuracy of the approximate interrelationships deduced in this paper, which is evidenced by the good matching between the experimental and fitting curves.

Figure 10 shows the evaluation of the relaxation modulus fitted by the approximation given by the EPL model to the data expressed by the SS and GLS expressions. The EPL coefficients are summarized in Table 7.

The results illustrated in Figure 10 show a close match between Sigmoid and EPL models, meaning that the proposed model yields a high predictive accuracy. Furthermore, the EPL models show their ability and easiness to fit and simulate the creep and relaxation behavior of polymeric materials.

## 8. Conclusions

This paper first presented a brief review of some methods of interconversion between linear viscoelastic properties of polymeric composites in the time domain. Their theoretical background was presented, and the relation between creep and relaxation and the constitutive integrals of linear viscoelasticity was summarized. It was shown that most of the classical methods of interconversion are based on the power-law and exponential methods of fitting. Despite their ease of application, these methods are not efficient in representing the full-path curve of the creep behavior of polymers and polymer composites, namely their three creep stages.

In the second part of the paper, a new approximate interconversion scheme that uses a convenient Laplace transform and an approximated Gamma function to convert the measured creep compliance to a relaxation modulus was presented. Data available in the literature were used to assess the consistency of the viscoelastic functions via the new interconversions presented in this work. The exponential-power-law (EPL) method of creep fitting (based on [13,14] and described in [15]) was converted to the EPL relaxation modulus expression, and the creep data provided in [23] were successfully fitted. To validate the EPL interconversion, the relaxation models identified in [27] were used as a reference. The results obtained show that the EPL fitting method proposed herein is very simple to implement and quick to apply, due to the small number of coefficients that need to be determined. In addition, the method is stable and accurate, as attested by the achieved Pearson correlation coefficient, greater than 0.97 for all test data considered.

## Figures and Tables

**Figure 1 polymers-12-03001-f001:**
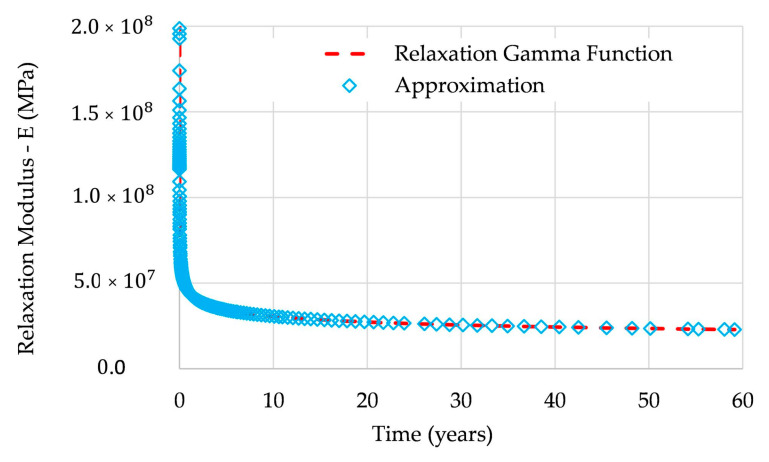
Exponential-power-law (EPL) Γ-function and approximation are given by (31).

**Figure 2 polymers-12-03001-f002:**
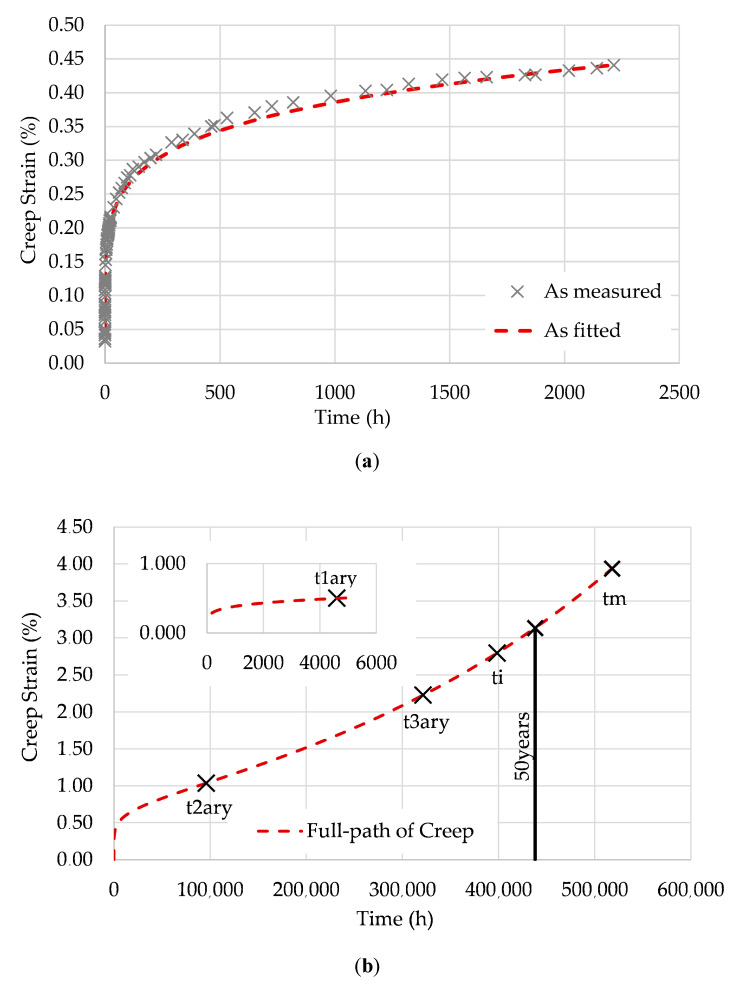
T28-S15: (**a**) data fitting; (**b**) full path of creep and transition points.

**Figure 3 polymers-12-03001-f003:**
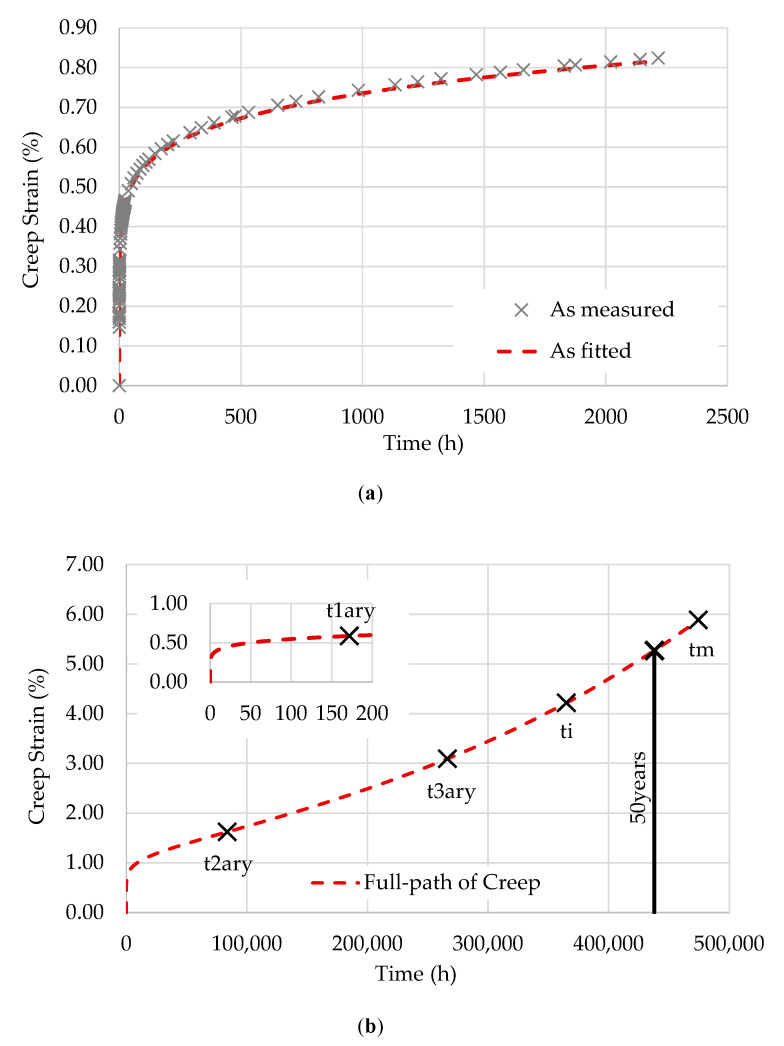
T28-S25: (**a**) data fitting; (**b**) full path of creep and transition points.

**Figure 4 polymers-12-03001-f004:**
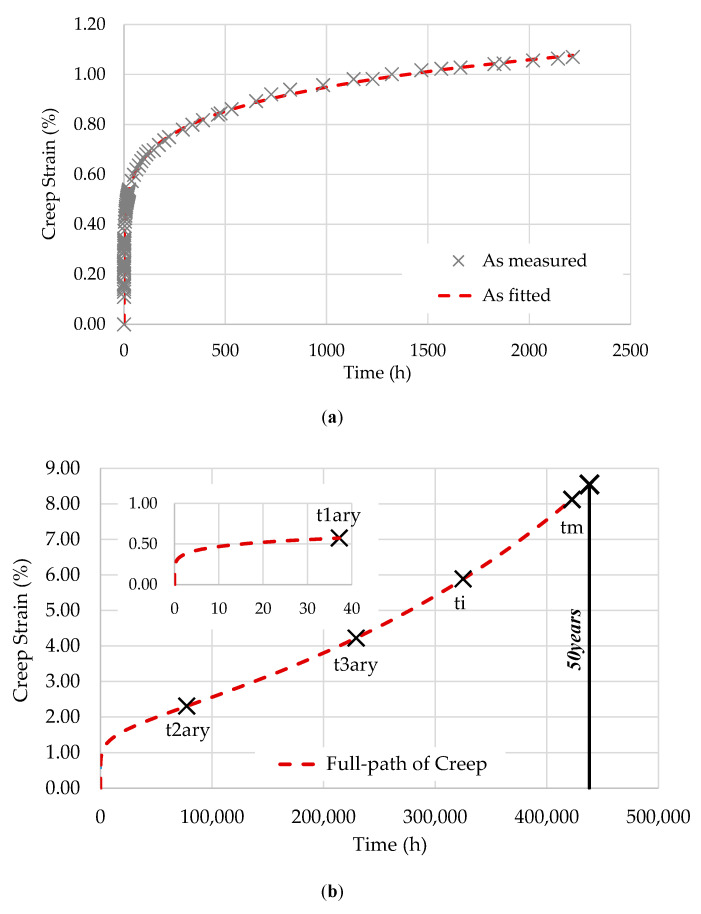
T28-S35: (**a**) data fitting; (**b**) full path of creep and transition points.

**Figure 5 polymers-12-03001-f005:**
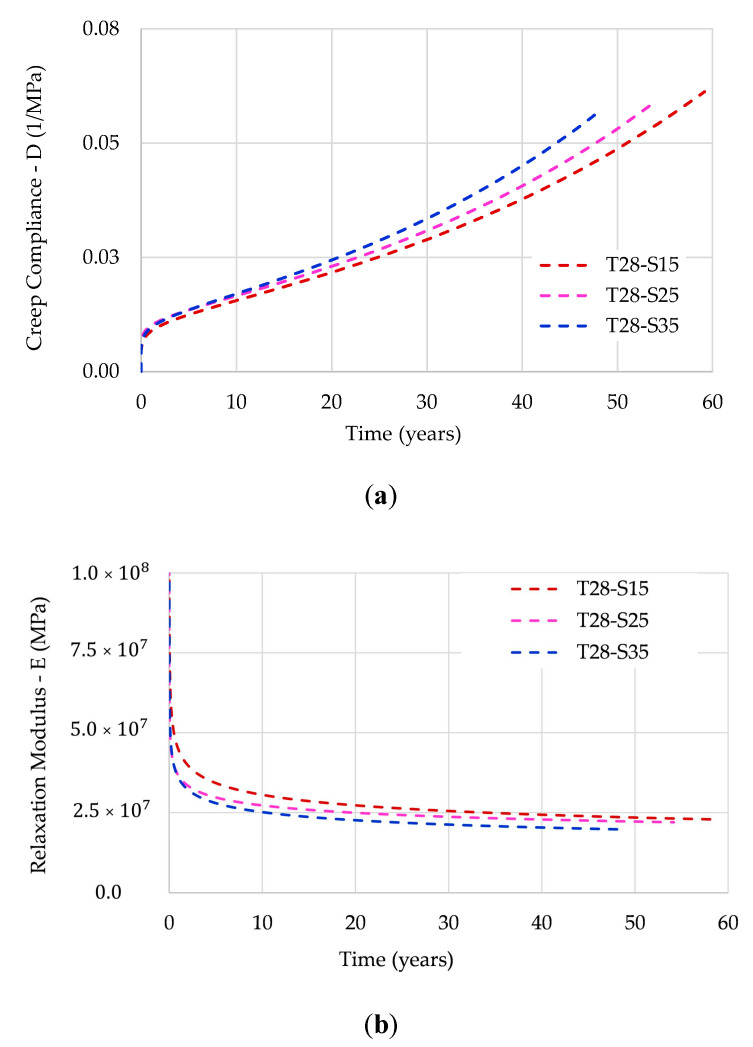
Full path of: (**a**) creep compliance; (**b**) relaxation modulus.

**Figure 6 polymers-12-03001-f006:**
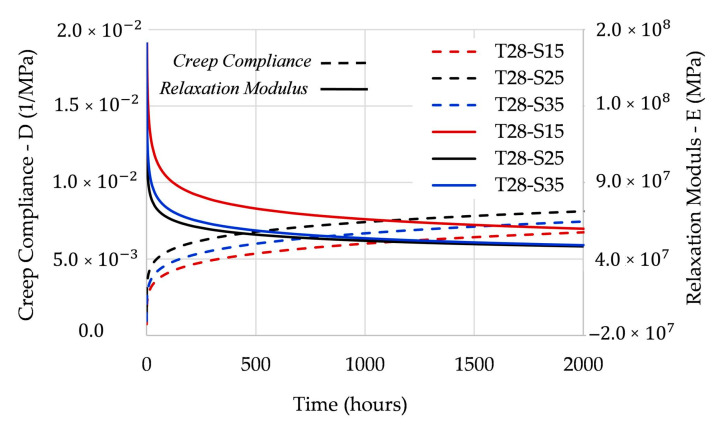
Relationship between the viscoelastic properties.

**Figure 7 polymers-12-03001-f007:**
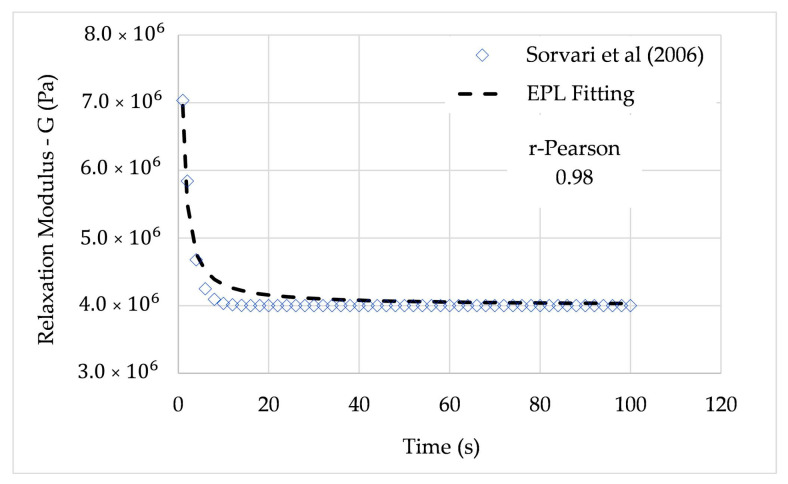
Relaxation modulus from Sorvari et al. (2006) and EPL fitted curve (λ=2 s).

**Figure 8 polymers-12-03001-f008:**
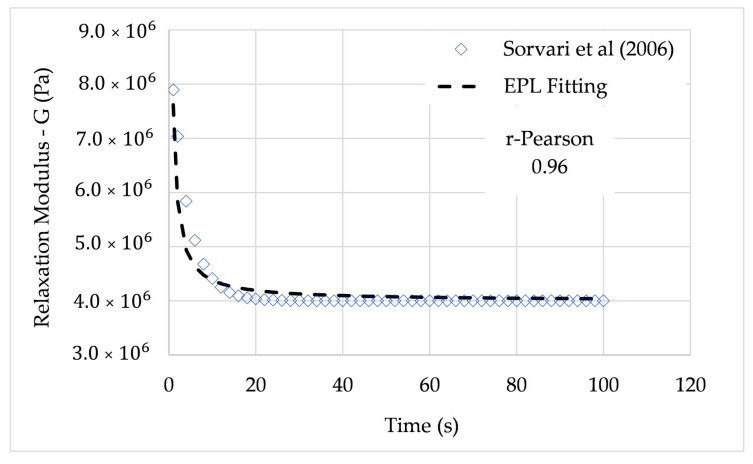
Relaxation modulus from Sorvari et al. (2006) and EPL fitted curve (λ=4 s).

**Figure 9 polymers-12-03001-f009:**
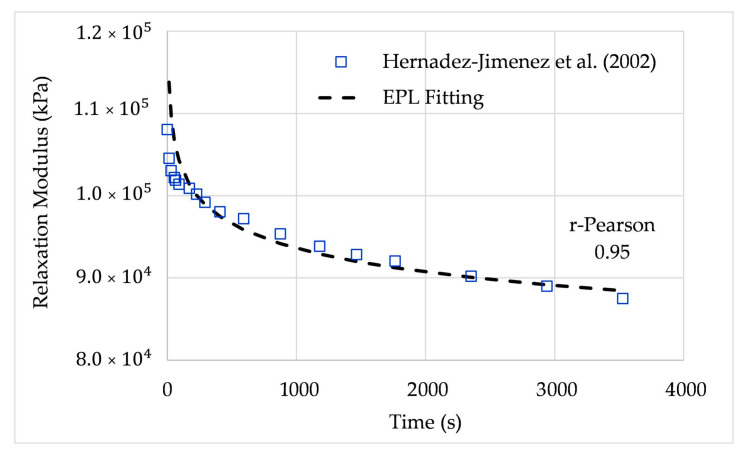
Relaxation modulus from Hernández-Jiménez et al. (2002) and EPL fitted curve for PMMA.

**Figure 10 polymers-12-03001-f010:**
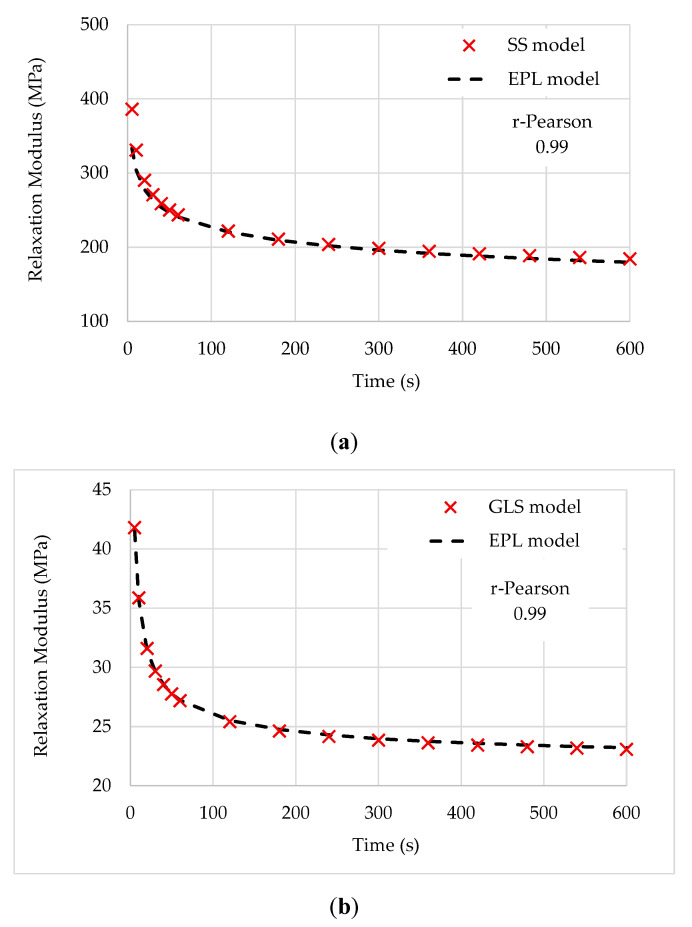
Relaxation modulus: (**a**) standard sigmoid (SS) x EPL models; (**b**) generalized logistic sigmoid (GLS) x EPL models.

**Table 1 polymers-12-03001-t001:** Laplace transform properties.

f(t)	$\hat{f}\left(t\right)$
$e^{\alpha t}$	$\frac{1}{s-\alpha }$
$t^{n},\;n>-1$	$\frac{n!}{s^{n+1}}$

**Table 2 polymers-12-03001-t002:** EPL coefficients.

Specimen	Stress	k	β	*t_i_*
	(MPa)			(h)
T28-S15	64.30	0.124	0.164	398,500
T28-S25	99.20	0.305	0.127	365,000
T28-S35	142.30	0.329	0.153	344,922

**Table 3 polymers-12-03001-t003:** Parameters to obtain the creep compliance by EPL expression.

Specimen	Stress	$D_{0}$	*β*	*t_i_*
	(MPa)	(1/MPa)		(h)
T28-S15	64.30	0.00193	0.164	398,500
T28-S25	99.20	0.00307	0.127	365,000
T28-S35	142.30	0.00231	0.153	344,922

**Table 4 polymers-12-03001-t004:** Parameters to obtain the creep compliance by EPL expression.

Specimen	Stress	$E_{0}$	*β*	*t_i_*
	(MPa)	(MPa)		(h)
T28-S15	64.30	518.55	0.164	398,500
T28-S25	99.20	325.25	0.127	365,000
T28-S35	142.30	432.52	0.153	344,922

**Table 5 polymers-12-03001-t005:** Parameters of the relaxation modulus by EPL expression.

Relaxation Time-*λ*	*k*	*β*	*t_i_*
[s]			[s]
2	7	0.980	100
4	5	0.982	97

**Table 6 polymers-12-03001-t006:** Parameters of the relaxation modulus by EPL expression.

*k*	*β*	*t_i_*
		(s)
0.0515	0.045	6600

**Table 7 polymers-12-03001-t007:** Parameters to obtain relaxation modulus by EPL expression.

Method	*k*	*β*	*t_i_*
			[h]
SS	0.910	0.140	3.70 × 10^2^
GLS	12.500	0.495	8.70 × 10^2^

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
