# Peer review of "A Combined Exponential-Power-Law Method for Interconversion between Viscoelastic Functions of Polymers and Polymer-Based Materials"

_polymers, 2020, doi:10.3390/polym12123001_

Round 1

Reviewer 1 Report

In the submitted manuscript, a new interconversion law between creep compliance and relaxation modulus for viscoelastic materials is presented and applied to model different experimental data from the literature. The manuscript structure is quite unusual, since the first part of the paper is a brief review of already published interconversion methods while the second part is devoted to the description of the new model and to its application and validation. In my opinion the manuscript deserves to be published in Polymers, but I suggest to the Authors to add some additional sentences in the Introduction part, aiming to better explain the importance of interconversion methods in predicting the viscoelastic behavior of polymers and polymer-based systems.

Reviewer 2 Report

The authors have performed an original modelling study about interconversion of viscoelastic properties in creep testing. I have enjoyed the reading. However, I think that this manuscript does not deserve to be published in Polymers unless one major change is addressed.

I understand that they have showed only the data that fit with their model. However, these data are coming only from two references (21 and 23). The authors should explain why only these two references are used to validate their model. In my opinion, two cases are not enough unless there is a strong justification.

In addtion, there are some minor changes to mention:

  • references 8 and 9 appear after 10 and 11.
  • the abstract is plenty of strange hyphens.
  • English should be revised (minor mistakes like known in line 150).

Round 2

Reviewer 2 Report

After the addition of the new reference and comparison with their model, I recomend this article to be accepted for publication in Polymers.